# G2S3: A gene graph-based imputation method for single-cell RNA sequencing data

**Weimiao Wu**[1], **Yunqing Liu**[1], **Qile Dai**[1], **Xiting Yan**[1,2]*, **Zuoheng Wang**[1]*

**1** Department of Biostatistics, Yale School of Public Health, New Haven, Connecticut, United States of America, **2** Section of Pulmonary, Critical Care and Sleep Medicine, Yale School of Medicine, New Haven, Connecticut, United States of America

* xiting.yan@yale.edu (XY); zuoheng.wang@yale.edu (ZW)

**Data Availability Statement:** G2S3 is an open-source MATLAB package that is freely available on GitHub https://github.com/ZWang-Lab/G2S3 under the MIT license. All data used in the publication is sourced from publicly available previously

## Abstract

Single-cell RNA sequencing technology provides an opportunity to study gene expression at single-cell resolution. However, prevalent dropout events result in high data sparsity and noise that may obscure downstream analyses in single-cell transcriptomic studies. We propose a new method, G2S3, that imputes dropouts by borrowing information from adjacent genes in a sparse gene graph learned from gene expression profiles across cells. We applied G2S3 and ten existing imputation methods to eight single-cell transcriptomic datasets and compared their performance. Our results demonstrated that G2S3 has superior overall performance in recovering gene expression, identifying cell subtypes, reconstructing cell trajectories, identifying differentially expressed genes, and recovering gene regulatory and correlation relationships. Moreover, G2S3 is computationally efficient for imputation in large-scale single-cell transcriptomic datasets.

## Author summary

Single-cell RNA sequencing (scRNA-seq) measures the expression profiles of individual cells. However, dropouts lead to an excessive number of zeros or close to zero values in the data, which may obscure downstream analyses. In this study, we developed G2S3, an imputation method that recovers gene expression in scRNA-seq data by borrowing information from adjacent genes in a gene graph learned by graph signal processing. G2S3 was shown to have superior performance in improving data quality. Moreover, G2S3 is computationally efficient in large-scale scRNA-seq data imputation.

This is a *PLOS Computational Biology* Methods paper.

## Introduction

Singe-cell RNA sequencing (scRNA-seq) has emerged as a state-of-the-art technique for transcriptome analysis. Compared to bulk RNA-seq that measures the average gene expression

published data. The detailed list of data sets used in the study is described in the "Real datasets" section. The code to reproduce all the analyses presented in the paper are available on GitHub https://github.com/ZWang-Lab/G2S3_paper2020.

**Funding:** WW, YL and ZW were supported by the National Institutes of Health (K01AA023321; https://www.nih.gov) and the National Science Foundation (DMS1916246; https://www.nsf.gov). YL and XY were supported by the National Institutes of Health (R21LM012884; https://www.nih.gov). The funders had no role in study design, data collection and analysis, decision to publish, or preparation of the manuscript.

**Competing interests:** The authors have declared that no competing interests exist.

profile of a mixed cell population, scRNA-seq measures the expression profile of individual cells and thus describes cell-to-cell stochasticity in gene expression. Applications of this technology in humans have revealed rare and novel cell types [1–3], cell population composition changes [4], and cell-type specific transcriptomic changes [3,5] that are associated with diseases. These findings have great potential to promote our understanding of cell function, disease pathogenesis, and treatment response for more precise therapeutic development [6,7]. However, analysis of scRNA-seq data can be challenging due to low library size, high noise level, and prevalent dropout events [8]. Particularly, dropouts lead to an excessive number of zeros or close to zero values in the data, especially for genes with low or moderate expression. These inaccurately measured gene expression levels may obscure downstream quantitative analyses such as cell clustering and differential expression analyses [6].

In the past few years, several imputation methods have been developed to recover dropout events in scRNA-seq data. A group of methods, including kNN-smoothing [9], MAGIC [10], scImpute [11], drImpute [12], and VIPER [13], assess between-cell similarity and impute dropouts in each cell using its similar cells. Specifically, kNN-smoothing uses step-wise k-nearest neighbors to aggregate information from the $k$ closest neighboring cells of each cell for imputation. MAGIC constructs an affinity matrix of cells and aggregates gene expression across similar cells via data diffusion to impute gene expression for each cell [10]. scImpute infers dropout events based on the dropout probability estimated from a Gamma-Gaussian mixture model and only imputes these events by borrowing information from similar cells within cell clusters detected by spectral clustering [11]. drImpute identifies similar cells through K-means clustering and performs imputation by averaging expression levels of cells within the same cluster [12]. While these imputation methods improved the quality of scRNA-seq data to some extent, they were found to eliminate natural cell-to-cell stochasticity which is an important piece of information available in scRNA-seq data compared to bulk RNA-seq data [13]. VIPER overcomes this limit by considering a sparse set of neighboring cells for imputation to preserve variation in gene expression across cells [13]. In general, imputation methods that borrow information across similar cells tend to intensify subject variation in scRNA-seq datasets with multiple subjects, which causes cells from the same subject to be more similar than those from different subjects. To address this issue, SAVER borrows information across similar genes instead of cells to impute gene expression using a penalized regression model [14]. There are other methods that leverage information from both genes and cells. For example, ALRA imputes gene expression using low-rank matrix approximation [15], and scTSSR uses two-side sparse self-representation matrices to capture gene-to-gene and cell-to-cell similarities for imputation [16]. In addition, machine learning-based methods, such as autoImpute [17], DCA [18], deepImpute [19] and SAUCIE [20], use deep neural network to impute dropout events. While computationally more efficient, these methods were found to generate false-positive results in differential expression analyses [21]. Recently, an ensemble approach, EnImpute, was developed to integrate results from multiple imputation methods using weighted trimmed mean [22].

In this article, we develop Sparse Gene Graph of Smooth Signals (G2S3), a gene graph-based method that imputes dropout events in scRNA-seq data by borrowing information across similar genes. G2S3 learns a sparse graph representation of gene-gene relationships from the data, in which each node represents a gene and is associated with a vector of expression levels in all cells considered as a signal on the graph. The graph is then optimized under the assumption that signals change smoothly between connected genes. Based on this graph, a transition matrix for a random walk is constructed so that the transition probabilities are higher between genes with similar expression levels across cells. A random walk on this graph imputes the expression level of each gene using the weighted average of expression levels from

itself and adjacent genes in the graph. In this way, G2S3, like SAVER, makes use of gene-gene relationships to recover the expression levels. However, unlike SAVER which uses a penalized regression model for imputation, G2S3 optimizes the gene graph structure using graph signal processing that captures nonlinear correlations among genes. The computational complexity of the G2S3 algorithm is a polynomial of the total number of genes in the graph, so it is computationally efficient, especially for large scRNA-seq datasets with hundreds of thousands of cells.

## Results

### Datasets and evaluation overview

We evaluated and compared the performance of G2S3 and ten existing imputation methods, SAVER, kNN-smoothing, MAGIC, scImpute, VIPER, ALRA, scTSSR, DCA, SAUCIE and EnImpute, in terms of (1) expression data recovery, (2) cell subtype separation, (3) cell trajectory inference, (4) differential gene identification, and (5) gene-gene relationship recovery. We applied these methods to eight scRNA-seq datasets that can be classified into five categories corresponding to the five criteria described above. The first category includes three unique molecular identifier (UMI)-based datasets in which down-sampling was performed to assess the method performance in recovering gene expression. These datasets are the Reyfman dataset from human lung tissue [23], the peripheral blood mononuclear cell (PBMC) dataset from human peripheral blood [24], and the Zeisel dataset from mouse cortex and hippocampus [25]. The second category was used to evaluate the method performance in separating different cell types and includes the Chu dataset of human embryonic stem (ES) cell-derived lineage-specific progenitors from seven known cell subtypes [26]. The third category was used to reconstruct cell trajectory and includes the Petropoulos dataset of cells from human preimplantation embryos collected on different embryonic days [27]. The fourth category was chosen to evaluate the method performance in identifying differentially expressed genes. It includes the Chu dataset, which is also included in the second category, and the Trapnell dataset of differentiating human myoblasts [28]. The last category includes two datasets to evaluate the method performance in recovering gene regulatory and correlation relationships among known regulators and marker genes. These datasets are the Paul dataset that contains a set of well-known transcriptional regulators of myeloid progenitor populations [29] and the Buettner dataset that contains 67 periodic marker genes whose expression level varies over cell cycle [30]. Table 1 summarizes the main features of all eight datasets. A more detailed description of these datasets is provided in the "Real datasets" section.

### Hyperparameter tuning in G2S3

The G2S3 algorithm used graph signal processing to learn a gene graph and performed a $t$-step random walk to borrow information from neighboring genes for imputation. The optimal value of the hyperparameter $t$ was selected by minimizing the mean squared error (MSE) between the imputed and observed data. We performed down-sampling on each dataset from the first category (Reyfman, PBMC and Zeisel) and evaluated the MSE as well as the gene-wise and cell-wise correlations of the G2S3 imputed data with reference data, for $t = 1, \ldots, 10$. S1 Fig shows the coefficient of variation (CV) of gene expression before and after down-sampling. In all datasets, although the CV of gene expression increased slightly after down-sampling, the correlation between the CV before and after down-sampling was 0.79 or higher, demonstrating that the down-sampled data well preserved the mean-variance relationship in the reference data. S2A Fig shows that the optimal value of $t$ is 1 in all three datasets based on the minimization of MSE. In addition, the one-step random walk in G2S3 achieved the greatest gene-wise

**Table 1. Detailed information on the eight scRNA-seq datasets used to compare the performance of imputation methods.**

| Experiment Category | Dataset | # Cells | Sample Type | Organism | Technique | UMI | Accession |
|---|---|---|---|---|---|---|---|
| Expression data recovery | Reyfman [23] | 5,437 | Lung tissue | Homo Sapiens | Drop-seq | Yes | GEO (GSE122960) |
| | PBMC [24] | 7,865 | Peripheral blood mononuclear cells | Homo Sapiens | Drop-seq | Yes | 10x Genomics* |
| | Zeisel [25] | 3,005 | Brain tissue | Mus Musculus | Drop-seq | Yes | Zeisel et al. [25] |
| Cell subtype separation | Chu [26] | 1,018 | Embryonic stem cells | Homo Sapiens | Fluidigm C1 | No | GEO (GSE75748) |
| Cell trajectory inference | Petropoulos [27] | 1,529 | Preimplantation embryos | Homo Sapiens | Smart-seq2 | No | Petropoulos et al. [27] |
| Differential gene identification | Chu [26] | 1,018 | Embryonic stem cells | Homo Sapiens | Fluidigm C1 | No | GEO (GSE75748) |
| | Trapnell [28] | 372 | Myoblasts | Homo Sapiens | Fluidigm C1 | No | GEO (GSE52529) |
| Gene-gene relationship recovery | Paul [29] | 2,730 | Bone marrow myeloid progenitor | Mus Musculus | MARS-seq | Yes | Paul et al. [29] |
| | Buettner [30] | 288 | Staged embryonic cells | Mus Musculus | Fluidigm C1 | No | ArrayExpress (E-MTAB-2805) |

* URL to access the dataset: https://support.10xgenomics.com/single-cell-gene-expression/datasets

and cell-wise correlations with the reference data (S2B Fig). This optimal choice of *t* was consistent with the hyperparameter selected by another diffusion-based imputation method [31].

## Expression data recovery in down-sampled datasets

We used three down-sampled datasets from the first category (Reyfman, PBMC and Zeisel) to assess the performance of all eleven imputation methods in recovering gene expression. Fig 1 shows the gene-wise Pearson correlation and cell-wise Spearman correlation between the imputed and reference data from each dataset. The correlation between the observed data without imputation and reference data was set as a benchmark. In all datasets, G2S3 consistently achieved the highest correlation with the reference data at both gene and cell levels; SAVER and scTSSR had slightly worse performance. EnImpute had comparable performance to G2S3 based on the cell-wise correlation but performed worse than G2S3, SAVER and scTSSR based on the gene-wise correlation. VIPER performed well in the Reyfman and PBMC datasets but not in the Zeisel dataset based on the gene-wise correlation, although the cell-wise correlations were much lower than G2S3, SAVER, scTSSR and EnImpute in all datasets. The other methods, kNN-smoothing, MAGIC, scImpute, ALRA and DCA, did not have comparable performance, especially based on the gene-wise correlation. SAUCIE did not have comparable performance to the other methods in all datasets (S3 Fig). To quantify the performance improvement of G2S3, one-sided t-test was applied to compare the gene-wise and cell-wise correlations of G2S3 to those of the other methods. G2S3 had significantly higher correlations than all the other methods across three datasets for both gene-wise and cell-wise correlations (p<0.05, S1 Table). Since genes with higher expression tend to have a lower dropout rate, they are usually easier to impute and have less imputation need than those with lower expression [8]. To demonstrate the impact of expression level on the method performance, we stratified genes into three subsets based on the proportion of cells expressing them in the down-sampled data: widely expressed (>80%, n = 155, 111, 110, respectively), mildly expressed (30%-80%, n = 615, 357, 1,902, respectively), and rarely expressed (<30%, n = 3,148, 1,830, 1,617,

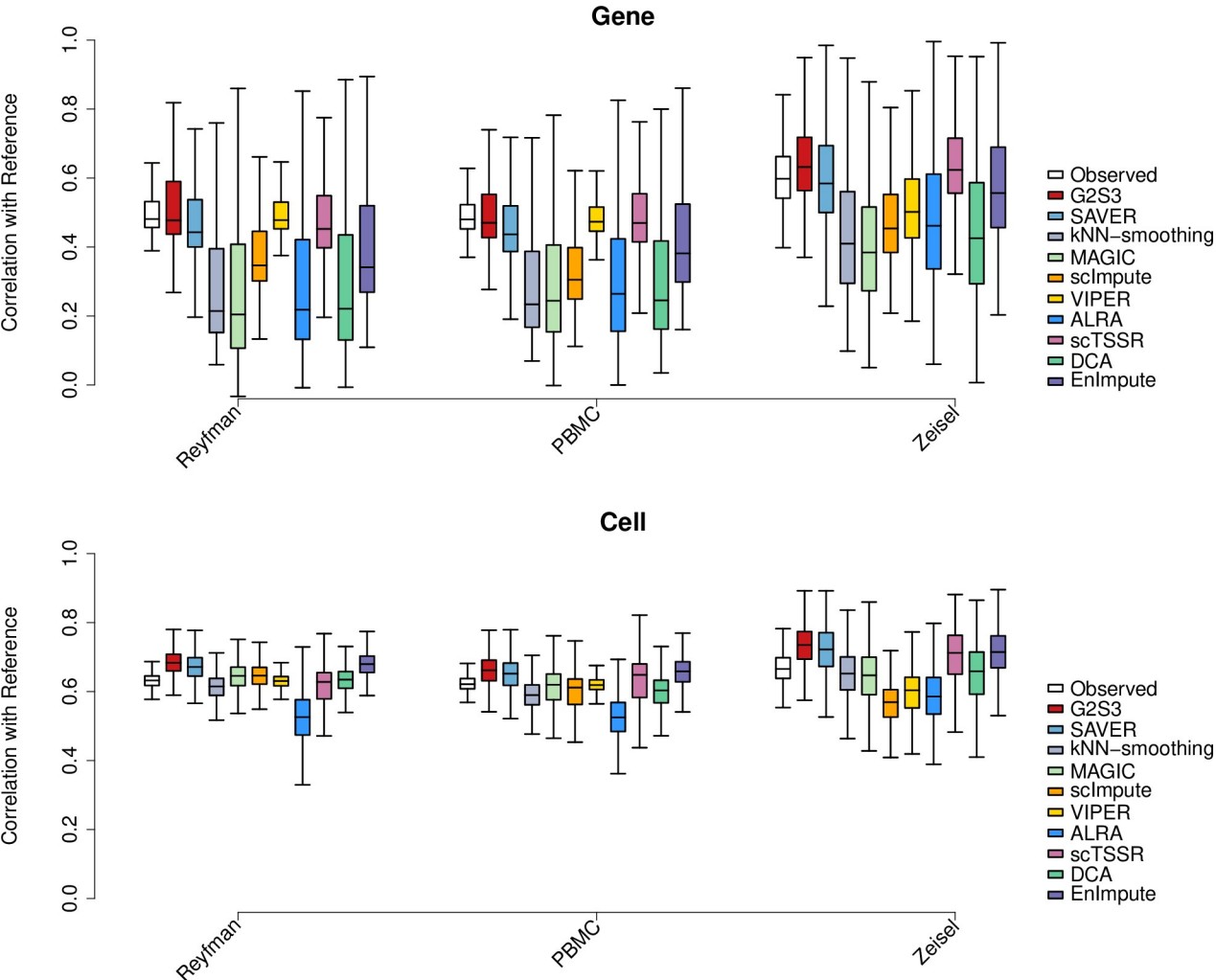

**Fig 1. Evaluation of expression data recovery of G2S3 by down-sampling.** Performance of imputation methods measured by correlation with reference data from the first category of datasets, using gene-wise (top) and cell-wise (bottom) correlation. Box plots show the median (center line), interquartile range (hinges), and 1.5 times the interquartile (whiskers).

respectively). S4 Fig shows the gene-wise and cell-wise correlations in each gene stratum. We can see that G2S3 improved both gene-wise and cell-wise correlations over the observed data for widely and mildly expressed genes. Moreover, G2S3 achieved the most superior recovery accuracy than the other methods for both widely and mildly expressed genes, although SAVER, scTSSR and EnImpute had comparable accuracy for widely expressed genes, suggesting the advantage of borrowing information from similar genes over from similar cells. For rarely expressed genes, all imputation methods did not improve the correlations compared to the observed data using both gene-wise and cell-wise correlation, suggesting that there is insufficient information for these genes to be successfully imputed. Overall, G2S3 provided the most accurate recovery of gene expression levels.

## Restoration of cell subtype separation

The second category of datasets was used to assess the performance of imputation methods in restoring separation between different cell types. In the Chu dataset, there were 7 cell types

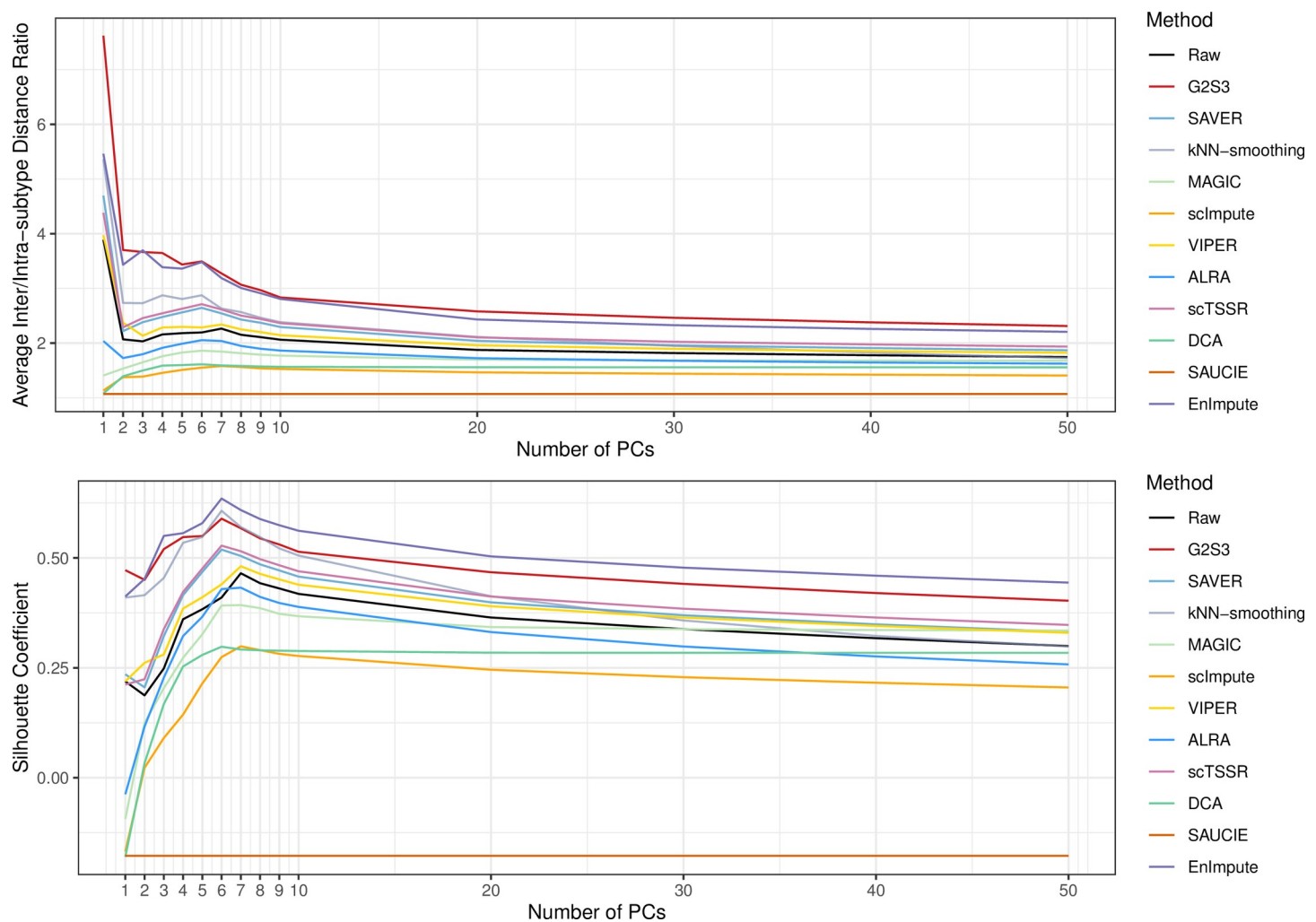

**Fig 2. Evaluation of G2S3 in improving cell subtype separation.** Average inter/intra-subtype distance ratio (top) and silhouette coefficient (bottom) to demonstrate cell subtype separation using the top principal components of the raw unimputed and imputed data by each method in the Chu dataset.

including two undifferentiated human ES cells (H1 and H9), human foreskin fibroblasts (HF), neuronal progenitor cells (NP), definitive endoderm cells (DE), endothelial cells (EC), and trophoblast-like cells (TB). To quantify the performance in separating these cell subtypes, we calculated the ratio of average inter-subtype distance to average intra-subtype distance using the top $K$ principal components (PCs) of the data before and after imputation, for $K = 1,\ldots,50$. We also calculated the silhouette coefficient that measures how similar cells are to cells from the same cell type compared to other cell types. In Fig 2, G2S3 and EnImpute had the highest inter/intra-subtype distance ratio and silhouette coefficient. Both methods performed better than the raw unimputed data, while MAGIC, scImpute, ALRA and DCA performed worse than the raw data. SAUCIE performed the worst. These results suggest that G2S3 greatly improved the separation between different cell types by enhancing the biologically meaningful information in the top PCs. Its performance was comparable to EnImpute, the ensemble method that takes advantage over several methods.

To demonstrate the comparison using cell clustering results, we generated uniform manifold approximation and projection (UMAP) plots in which cells were colored to represent the

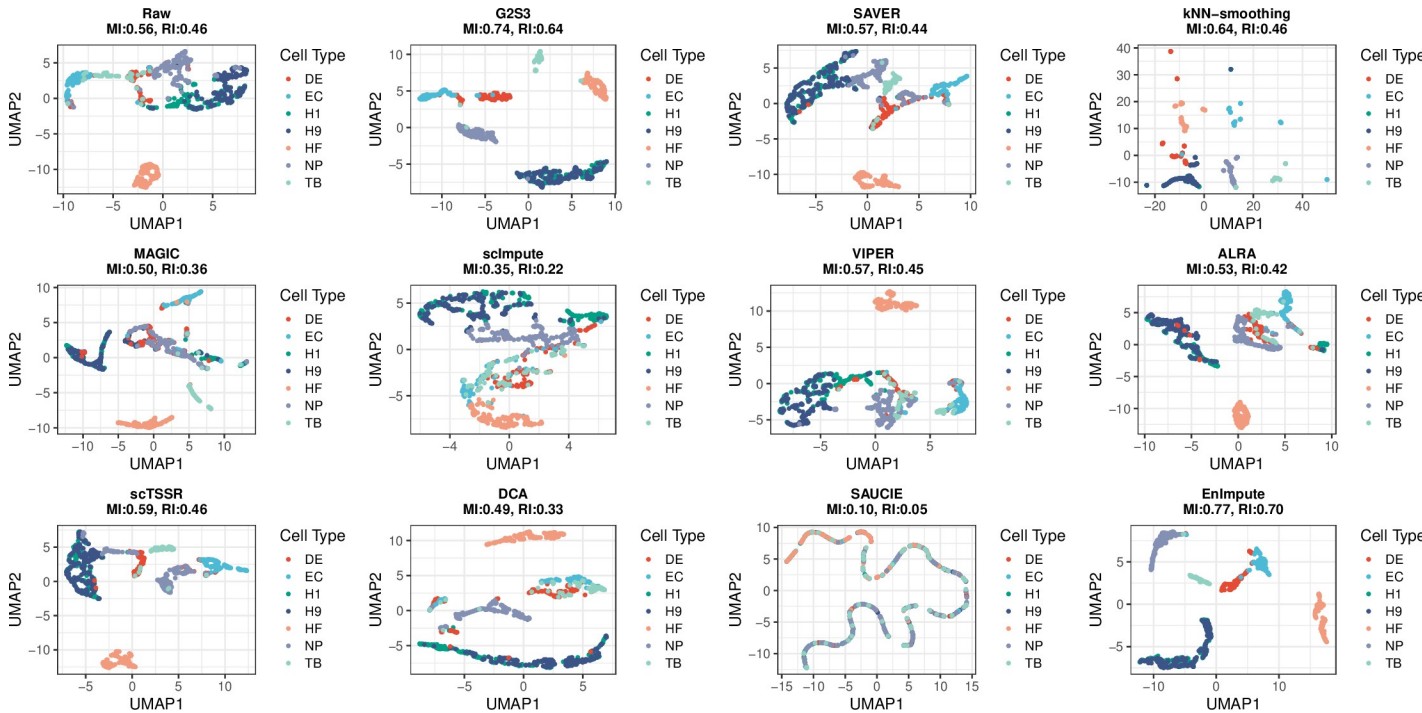

**Fig 3. Plots showing 2D-Visualization of the Chu dataset.** UMAP plots of the raw unimputed and imputed data by all methods. Cells are colored by true cell subtype labels. The normalized mutual information (MI) and adjusted rand index (RI) are calculated to measure the consistency between cell clustering results and true cell subtype labels.

seven cell types in the original dataset. The normalized mutual information (MI) and adjusted rand index (RI) were calculated to measure the consistency between cell clustering results and true cell subtype labels. Fig 3 shows that the imputed data by G2S3 and EnImpute had better separation of all cell subtypes than the raw unimputed data, except for H1 and H9 cells. Given that both H1 and H9 are undifferentiated human ES cells, it is expected that separating them is more difficult due to the relative homogeneity of human ES cells compared to the progenitors. In contrast, the other imputation methods did not have comparable improvement that some of which even reduced the separation of different cell types. Specifically, DE cells were mixed with EC and TB cells in the raw data and were not separated from the other cell subtypes by all methods except G2S3 and EnImpute. MAGIC was able to separate EC, HF and TB cells from each other and from the rest of the cell subtypes, while SAVER was able to separate EC and HF cells from each other and from the rest of the cell subtypes. VIPER, ALRA, scTSSR and DCA only separated HF cells from the rest, similar to the raw data. The imputed data by kNN-smoothing formed many small clusters. scImpute tended to mix different cell types into one cluster. SAUCIE overly smoothed the data and was not able to separate any cell types. Based on the two measures of consistency between cell clustering results and true cell subtype labels, EnImpute had the best separation of the cell subtypes (MI = 0.77, RI = 0.70) and G2S3 was the second best (MI = 0.74, RI = 0.64), while the other methods did not have comparable performance. Notice that EnImpute is an ensemble method that combines imputation results from multiple methods, and G2S3 is the only method that achieved comparable performance to EnImpute.

S5 Fig demonstrates the expression of two cell subtype marker genes *GATA6*, a marker gene of DE cells, and *NANOG*, a marker gene of H1/H9 cells [26], across all cells in the raw

unimputed and imputed data by all methods. The normalized MI and adjusted RI that measure the consistency between cell clustering results, based on these two marker genes and true cell labels for DE and H1/H9 cells, were also calculated. We can see that G2S3 provided the best separation between H1/H9 cells, DE cells and other cell subtypes. Specifically, while the raw data mixed H1/H9 cells with other cell subtypes, G2S3 successfully recovered the expression of *GATA6* and *NANOG* to better separate DE and H1/H9 cell subtypes both from each other and from the other cell subtypes. The cell clustering results on the G2S3 imputed data achieved the highest consistency with true cell subtype labels, indicating its best performance. None of the other methods had comparable performance. DCA separated H1/H9 cells but had DE cells marginally overlapped with other cell types. We observed many small clusters of cells after imputation by kNN-smoothing, similar to the pattern displayed in Fig 3. The other methods did not improve cell subtype separation compared to the raw data. In addition, the imputed data by VIPER, kNN-smoothing and ALRA still had a large proportion of dropout events. These results suggest that G2S3 had the best performance in restoring the separation of different cell types, preserving biological meaningful variations, and reducing technical noise.

## Improvement in cell trajectory inference

Reconstruction of cell trajectories using scRNA-seq data is important for investigating a dynamic process. However, dropout events may impair pseudo-time inference. We used the Petropoulos dataset to evaluate the performance of all imputation methods in cell trajectory inference. This dataset consists of human preimplantation embryonic cells from five embryonic days (E3-E7) that represent differentiation stage or age of the embryonic cells. We used Monocle 2 to infer pseudo-time from the raw unimputed and imputed data by each method [32] and compared this to the actual embryonic days of the cells for performance evaluation. The pseudotemporal ordering score (POS) and Kendall rank correlation coefficient (Cor) were calculated to measure the consistency. Fig 4 shows cell trajectories in the raw and imputed data by all methods. The cell trajectory plots showed the sequential layout of cells from earlier to later embryonic days. The imputed data by G2S3, scImpute, VIPER and EnImpute had the highest consistency with the actual embryonic days, indicating their best performance among all methods. SAVER, kNN-smoothing, MAGIC, ALRA and DCA formed the

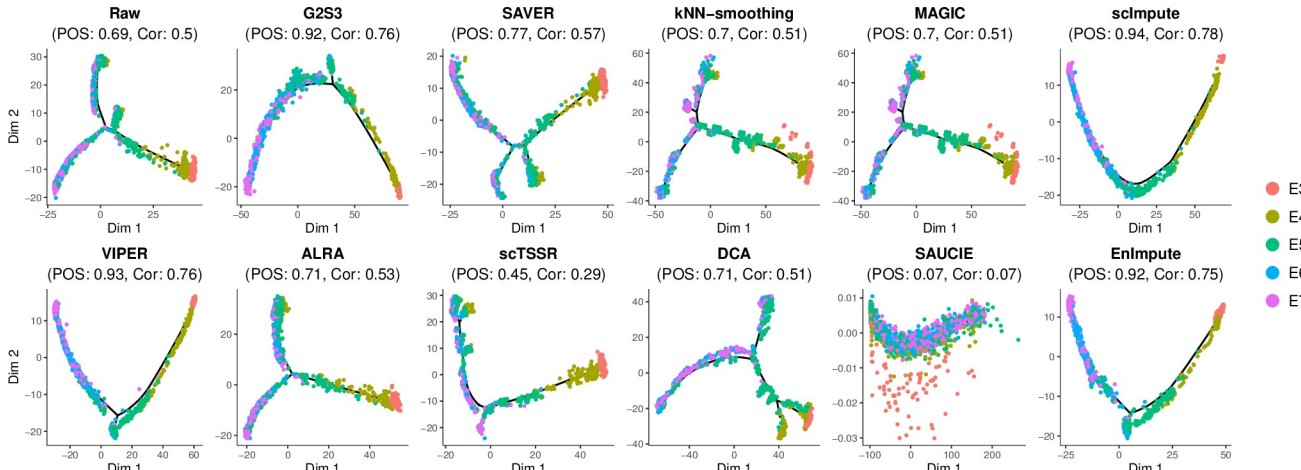

**Fig 4. Visualization of cell trajectories in the raw and imputed data by all methods.** Cells are projected into two-dimensional space using reversed graph embedding. Pseudotemporal ordering score (POS) and Kendall rank correlation coefficient (Cor) are used to measure the consistency between the actual embryonic days and the reconstructed pseudo-time.

second tier of methods with lower consistency. scTSSR performed worse than the raw data. SAUCIE had significantly lower consistency (POS = 0.07, Cor = 0.07) compared to all other methods in cell trajectory inference. Furthermore, the cell trajectory plots showed an increased heterogeneity among cells from later embryonic days, especially starting from embryonic day 5. This was consistent with the observation of a significant embryonic cell differentiation event on embryonic day 5 [27].

## Improvement in differential expression analysis

One common analytical task for scRNA-seq studies is to identify genes differentially expressed between cells from two groups of subjects or two cell types. In this section, we used two datasets to evaluate and compare the improvement in downstream differential expression analysis before and after imputation by all methods: the Chu dataset of different cell types and the Trapnell dataset of differentiating human myoblasts. Besides the scRNA-seq data, both datasets provide bulk RNA-seq data on the same samples with each sample consisting of cells from only one cell type. We expect that the differentially expressed genes identified from the bulk RNA-seq data overlap with that from the scRNA-seq data. Therefore, we treated the differentially expressed genes in the bulk RNA-seq data as ground truth and compared methods by assessing the prediction accuracy of the ground truth in the scRNA-seq data imputed by different methods using receiver operating characteristic (ROC) curves.

In the Chu dataset, we identified marker genes that differentiate the two cell types: H1 and NP cells. Fig 5A shows that G2S3 had the highest area under the curve (AUC) in detecting differentially expressed genes. kNN-smoothing, DCA and EnImpute had an AUC score lower than G2S3 but higher than the raw data. The other methods had comparable performance to the raw data except MAGIC, which had the lowest AUC. This is likely due to the fact that a small cluster of NP cells were mixed with H1 cells after imputation by MAGIC (Fig 3),

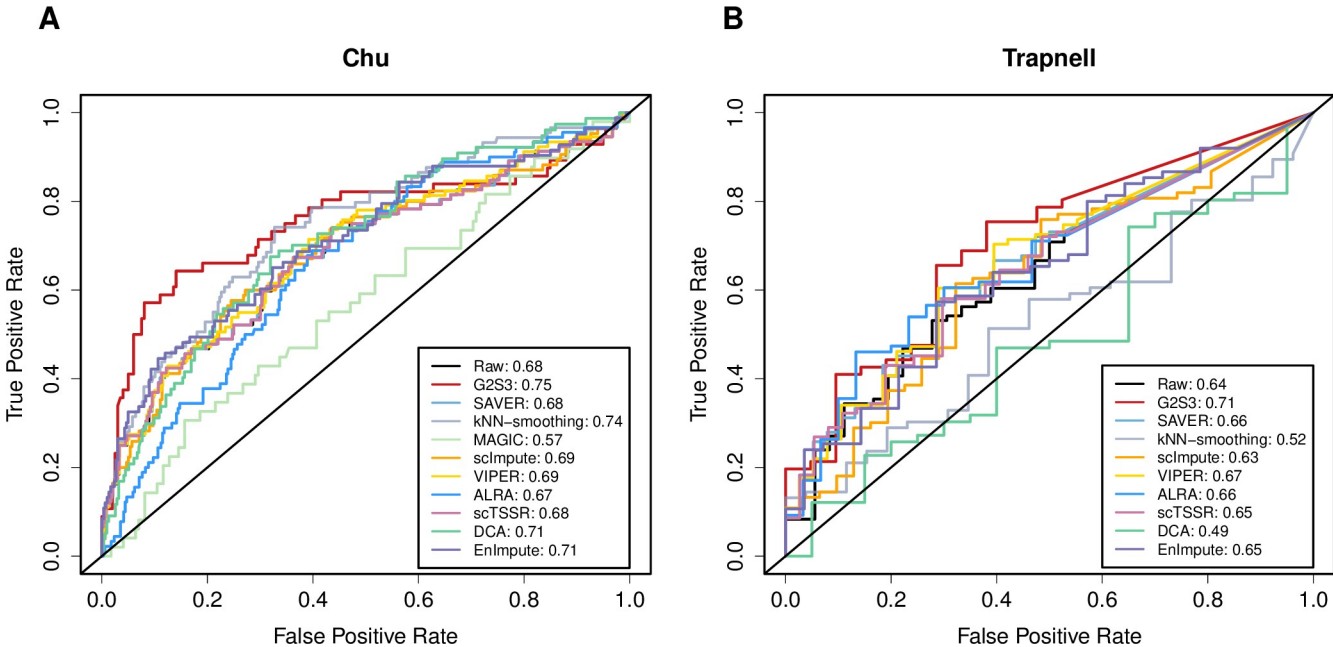

**Fig 5. Receiver operating characteristic (ROC) curves demonstrating improvement in differential expression analysis.** ROC curves measuring the prediction accuracy in scRNA-seq data on differentially expressed genes identified in bulk RNA-seq data from the same samples in the Chu (A) and Trapnell (B) datasets.

resulting in compromised performance in marker gene identification. Our results were largely consistent with a previous evaluation of imputation methods in identifying differentially expressed genes using Fluidigm C1 data [33]. No genes achieved significance in the imputed data by SAUCIE, so the result of SAUCIE could not be shown. DE cells had two or more sub-clusters in UMAP and one subcluster was mixed with EC cells (Fig 3). Similar to H1, H9 cells are undifferentiated human ES cells. To demonstrate results on comprehensive cell types, we further compared H1 cells with all other cell types except H9 and DE cells (S6 Fig). The results on three out of the four cell types compared to H1 cells demonstrated the best performance of G2S3. TB cells is the only cell type for which G2S3 did not achieve the best performance. All other methods, except EnImpute, did not achieve higher AUC than the raw data, indicating the lack of benefit by performing data imputation for genes differentially expressed between H1 and TB cells, regardless of imputation methods. In the Trapnell dataset, we performed differential expression analysis between undifferentiated primary human myoblasts and mature myotubes captured 72 hours after inducing differentiation. Fig 5B shows that G2S3 achieved the highest AUC indicating its best performance, followed by VIPER. kNN-smoothing and DCA had much worse performance than the raw data. No genes achieved significance in the imputed data by MAGIC and SAUCIE, so their results could not be shown. Altogether, the results from both datasets showed that G2S3 had the best improvement in the downstream differential expression analysis.

## Gene-gene relationship recovery

We compared the method performance in recovering gene regulatory and correlation relationships using two scRNA-seq dastasets. In the Paul dataset, we examined the regulatory relationships between ten well-known transcription factors in the development of blood cells before and after imputation [34]. In the Buettner dataset, we investigated the correlation among a set of 67 periodic marker genes before and after imputation, in which 16 genes have peak expression in the G1/S phase and 51 genes have peak expression in the G2/M phase [30].

In the Paul dataset, the regulatory relationships among the ten key regulators of the transcriptional differentiation of megakaryocyte/erythrocyte progenitors and granulocyte/macrophage progenitors in the raw data and the imputed data by each method were used for performance evaluation. The gene regulatory network (GRN) of these regulators was established in a previous study based on biological experiments [35–37] and served as the ground truth. We reconstructed GRNs using four methods, PIDC [38], GENIE3 [39], GRNBoost2 [40], and PPCOR [41], in the raw and imputed datasets. The inferred GRNs were compared to the ground-truth network to measure the prediction accuracy using the area under the receiver operating characteristic curve (AUROC) and the area under the precision-recall curve (AUPRC). For each imputation method, we reported the AUROC and AUPRC ratio (AUROC/AUPRC divided by that of a random predictor) with 50 replications. Fig 6 shows that G2S3 achieved the highest AUROC ratio in three out of the four GRN inference methods and performed slightly worse than scImpute using GENIE3. The prediction accuracy of scImpute was much lower than a random predictor using GRNBoost2 and PPCOR. The AUROC ratios of GRNs inferred from the imputed data by MAGIC and ALRA were either equal to or much lower than that from a random predictor, suggesting that the gene regulatory relationships were distorted after imputation. S7 Fig demonstrates the results based on the AUPRC ratio. G2S3 and kNN-smoothing had better prediction accuracy than other imputation methods in restoring gene regulatory relationships across all GRN inference methods.

We also examined the pairwise correlations between these ten key regulators. Based on previous studies [35–37], inhibitory and activatory gene pairs were defined, among which

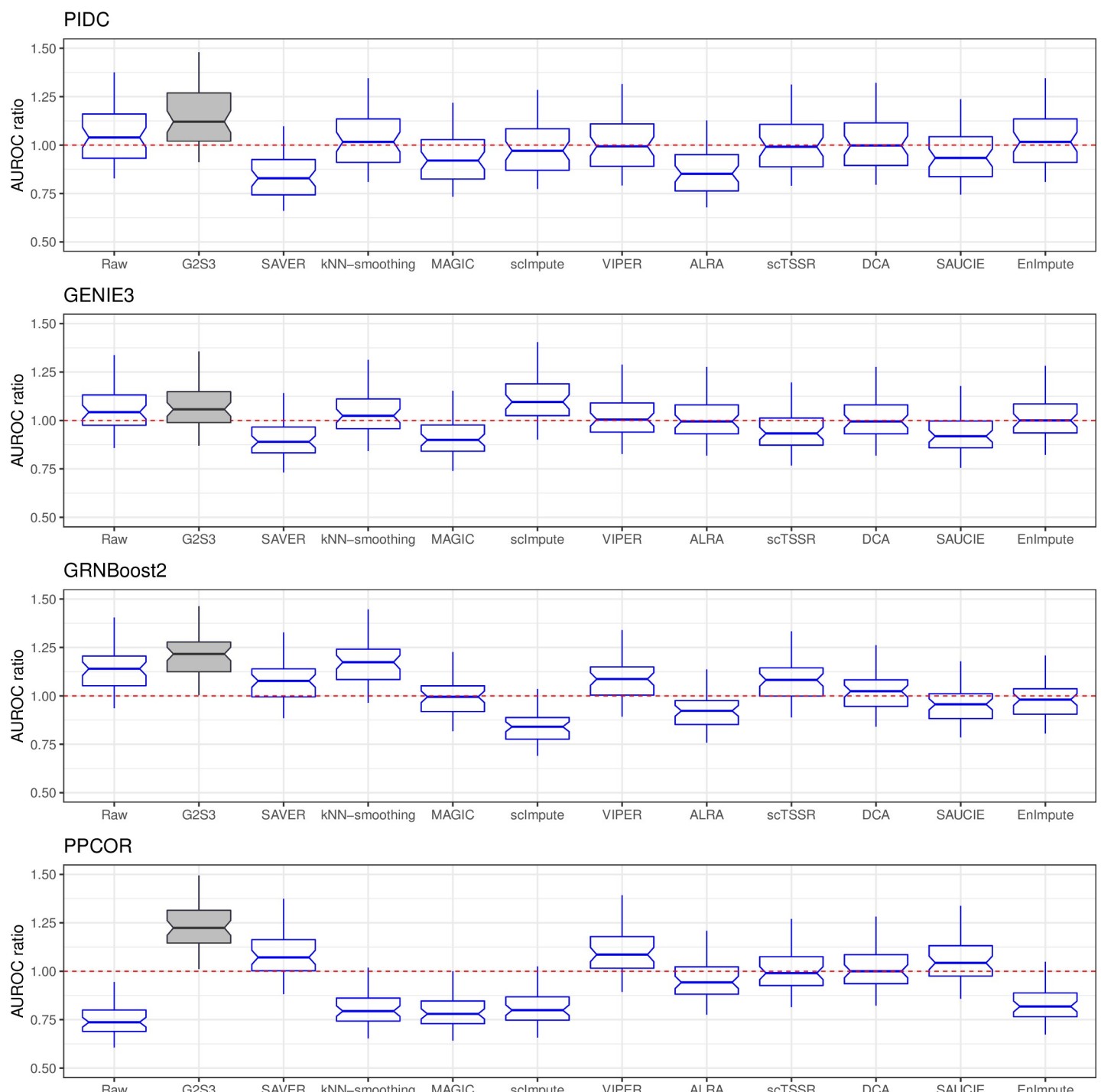

**Fig 6. Performance of G2S3 in recovering gene regulatory relationships.** Boxplots showing the area under the receiver operating characteristic curve (AUROC) ratios that measure the accuracy of inferred GRNs using the imputed data by different imputation methods. PIDC, GENIE3, GRNBoost2 and PPCOR are used to infer GRNs. Red line indicates the performance of a random predictor.

inhibitory pairs were expected to have negative correlation while activatory pairs were expected to have positive correlation. The mutually inhibitory pairs of genes include *Fli*1 vs. *Klf*1, *Egr*1 vs. *Gfi*1, *Cebpa* vs. *Gata*1, and *Sfpi*1 vs. *Gata*1; and the mutually activatory pairs

include *Sfpi*1 vs. *Cebpa*, *Zfpm*1 vs. *Gata*1, and *Klf*1 vs. *Gata*1. S8 Fig shows that most of the methods were able to enhance the pairwise correlations in the correct direction after imputation. Overall, G2S3 and SAVER showed the greatest enhancement of pairwise correlations for both inhibitory and activatory pairs, followed by kNN-smoothing and EnImpute. Although MAGIC intensified the pairwise correlations, most activatory pairs had correlations close to 1 after imputation. ALRA and DCA strengthened the pairwise correlations for activatory pairs but did not improve much for inhibitory pairs. Imputation by SAUCIE resulted in all gene pairs to be highly positively correlated. We further examined the correlation enhancement of each method by plotting all gene pairs (S9 and S10 Figs). While many methods, for example, G2S3, SAVER, kNN-smoothing, ALRA, DCA and EnImpute, had good performance in enhancing positive correlations, most of them had less satisfactory performance in negatively correlated gene pairs. For inhibitory gene pairs (S9 Fig), only G2S3 and SAVER displayed negatively correlated curves in which the expression level of one gene decreased with an increase of the other. kNN-smoothing, DCA and EnImpute tended to over-impute to the extent that only one gene was expressed in the same cell after imputation. This goes against the observation from the raw data and previous literature showing that the higher expression of one gene, the lower, rather than completely shutting off, the expression of the other. Additionally, SAUCIE imputed all mutually inhibitory gene pairs to be positively correlated. For activatory gene pairs (S10 Fig), most methods enhanced the pairwise correlations except scImpute and VIPER, which did not improve much compared to the raw data. In addition, the imputed data by MAGIC and SAUCIE formed a nearly straight diagonal line, suggesting that the imputed data was over-smoothed such that the cell-level biological variation was attenuated.

In the Buettner dataset, we expect pairs of periodic genes whose expression peak in the same phase of cell cycle to be positively correlated and those that peak during different phases to be negatively correlated. There are 67 marker genes for G1/S and G2/M phases [34]. We examined the correlation of all 2,211 marker gene pairs in the raw data and imputed data by each method. The proportion of gene pairs whose correlations are in the correct direction was used for performance comparison. Table 2 shows that all methods had comparable performance in maintaining a high proportion of positively correlated gene pairs, whereas their performance varied in restoring negatively correlated gene pairs. G2S3, SAVER and EnImpute were able to recover 28% or more of the negatively correlated gene pairs. All gene pairs became positively correlated after imputation by MAGIC, scImpute, VIPER, ALRA, DCA and

**Table 2. Fraction of periodic gene pairs with correct direction of correlation in the raw and imputed data by each method.**

| Imputation Methods | Positive Pairs | Negative Pairs |
|---|---|---|
| Raw | 1.00 | 0.00 |
| G2S3 | 0.91 | 0.32 |
| SAVER | 0.94 | 0.28 |
| kNN-smoothing | 0.97 | 0.17 |
| MAGIC | 1.00 | 0.00 |
| scImpute | 1.00 | 0.00 |
| VIPER | 1.00 | 0.00 |
| ALRA | 1.00 | 0.00 |
| scTSSR | 0.98 | 0.11 |
| DCA | 1.00 | 0.00 |
| SAUCIE | 1.00 | 0.00 |
| EnImpute | 0.91 | 0.46 |

SAUCIE, thus no negative correlation was observed after imputation. Similar observations were found in a previous study in which some of these methods introduced a large number of positive gene correlations after imputation, many of which may be spurious [14].

In summary, the results from both datasets suggested that G2S3 enhanced gene-gene relationships especially for negatively correlated gene pairs. In negatively correlated gene pairs, the expression of one gene is inhibited by the other, resulting in one of the genes being lowly expressed. In general, as genes with low expression are more difficult to impute, restoring negative correlation is thus a more challenging task for imputation.

## Summary of method performance

We evaluated and compared the performance of G2S3 and the other ten imputation methods using five evaluation criteria corresponding to five downstream analyses of scRNA-seq data. Fig 7 summarizes the overall performance of all methods. G2S3 was ranked first in three out of the five evaluation criteria, second in cell clustering, and third in cell trajectory inference. For those criteria under which G2S3 did not achieve the best performance, it had close or comparable performance to the best method. No other method achieved the best performance in as many criteria as G2S3. Overall, G2S3 performed the best among all the methods, followed by EnImpute, SAVER and VIPER.

## Computation time

While SAVER and EnImpute have comparable performance to G2S3 in some datasets, G2S3 is computationally more efficient (S2 Table). Since both G2S3 and SAVER are gene network-based imputation methods, their computation time is expected to increase with the number of genes to be imputed. This makes gene network-based methods more suitable than those based on cell similarity for large scRNA-seq datasets with tens or even hundreds of thousands of cells. In real data analysis, G2S3 was on average about twenty times faster than SAVER. EnImpute is an ensemble method that relies on imputation results from multiple methods and therefore is slower than SAVER. On the other hand, the computation time of imputation methods that borrow information from similar cells increases dramatically with the number of cells in the data. As demonstrated in a previous study, scImpute and VIPER were unable to scale beyond 10K cells within 24 hours [19]. In our assessment, VIPER took about two days to impute the down-sampled datasets with several thousands of genes, while other methods finished within several minutes.

## Discussion

We have developed a new method, G2S3, to impute dropouts in scRNA-seq data. G2S3 learns a sparse gene graph from scRNA-seq data and borrows information from nearby genes in the graph for imputation. We evaluated and compared the performance of G2S3 and ten existing imputation methods in terms of recovering gene expression, restoring cell subtype separation, reconstructing cell trajectories, identifying differentially expressed genes, and restoring gene regulatory and correlation relationships using eight scRNA-seq datasets. Overall comparison based on the five evaluation criteria showed that G2S3 achieved the best performance. Furthermore, G2S3 is computationally efficient for large-scale scRNA-seq data imputation.

Unlike imputation methods that borrow information across similar cells, G2S3 harnesses the structural relationship among genes obtained through graph signal processing to perform imputation. Using eight real datasets, we showed that methods relying on cell similarity tend to remove biological variation among cells and intensify subject-level batch effects. In contrast, G2S3 enhances cell subtype separation and thus relatively reduces variations in cells from the

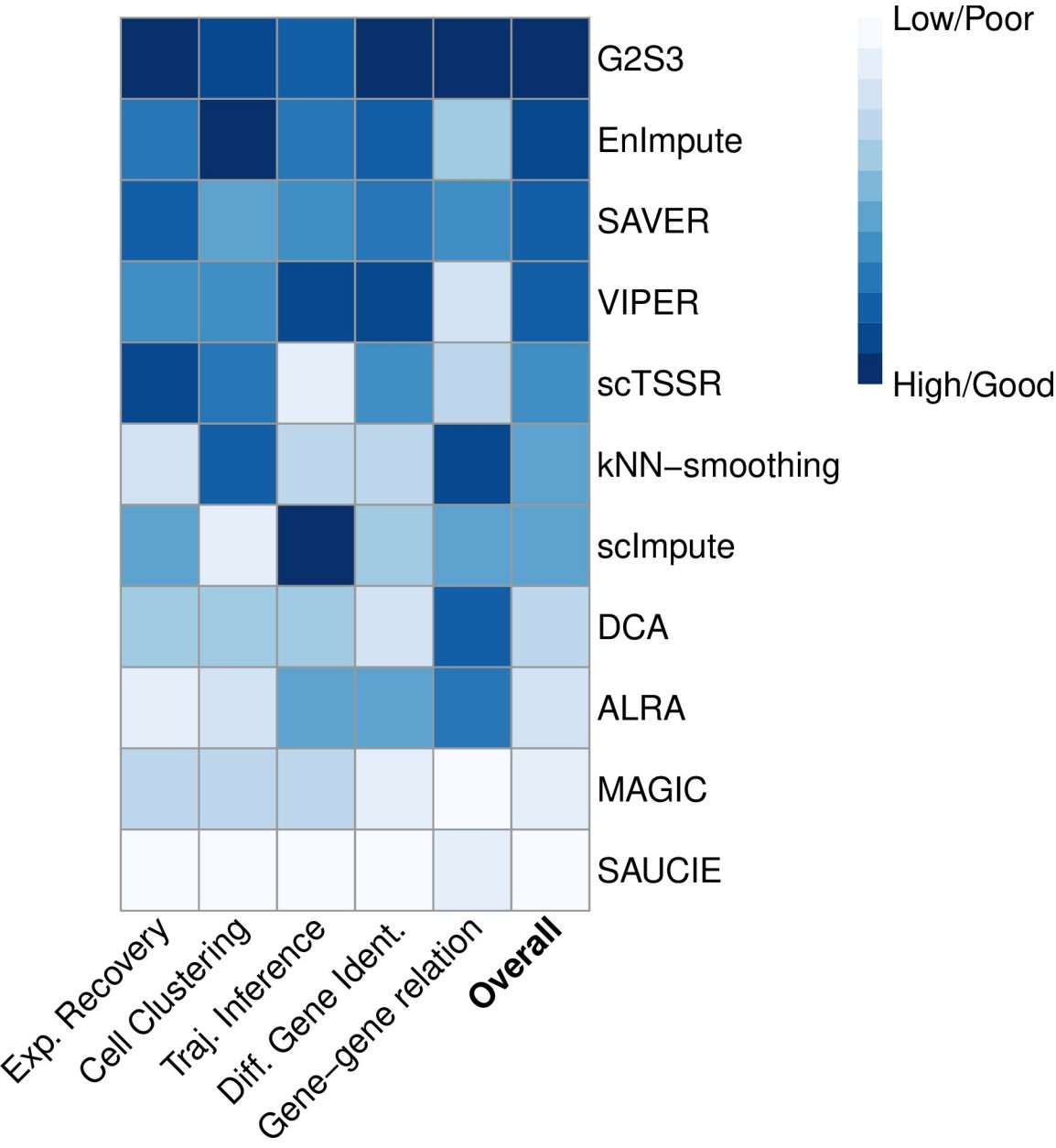

**Fig 7. Summary of performance of G2S3 and other imputation methods.** A heatmap demonstrating method performance based on the five evaluation criteria. The left five columns display performance rank using each of the five evaluation criteria. The rightmost column displays the overall performance rank based on the sum of the five ranks.

same cell type and subject. The down-sampling and differential expression analysis results showed that G2S3 outperformed the other methods. Of note, imputation methods such as SAVER, scImpute and VIPER used parametric models for gene expression. However, as the noise distribution varies across different scRNA-seq platforms, assumptions of the parametric models may be violated, particularly for new technologies. Graph signal processing extracts signals from data by optimizing a smoothness regulated objective function, so in principle, it is less sensitive to the noise distribution. To our knowledge, there are two imputation methods that use gene graph/network for imputation in scRNA-seq data, published during the

preparation of this manuscript: netNMF-sc [42] uses network-regularized non-negative matrix factorization to leverage gene-gene interactions for imputation; and netSmooth [43] incorporates protein-protein interaction networks to smooth gene expression values. Both methods require prior information on gene-gene interactions from RNA-seq or microarray studies of bulk tissue. In contrast, G2S3 learns gene network structure in an unbiased way from scRNA-seq data. In our experiments, G2S3 had comparable performance to EnImpute, an ensemble learning method that combines results from multiple imputation methods.

G2S3 learns gene-gene relationship by optimizing a sparse gene graph and at the same time allows expression levels to change smoothly between closely connected genes. Since many gene networks and biochemical networks are sparse [32,44,45], the sparsity property is important for inferring gene network. There are several methods available for constructing gene network, many of them are kernel-based, which result in full weight matrices where sparsity is to be imposed afterwards, for example, thresholding the adjacency weights. We found that the top eigenvectors of graph Laplacian on the gene networks learned from Gaussian kernel were highly correlated with dropout rate, suggesting that dropout events tend to bias the construction of gene network in scRNA-seq data. Based on our evaluation of the hyperparameter in G2S3, we chose to use a one-step random walk for datasets in this article to avoid oversmoothing, because multiple steps in a random walk tend to overly smooth the data and lead to compromised performance. Nevertheless, we implemented an MSE-based tuning on the number of steps in the algorithm. Similar observations were reported in a recent study discussing parameter tuning for diffusion-based imputation methods in scRNA-seq data [31]. It showed that for many diffusion-based methods including MAGIC, single step ($t = 1$) yielded better performance than multiple steps or iterations until convergence. For UMI-based datasets, to account for the effect of varying sequencing depths, we recommend normalizing UMI counts before applying G2S3 for accurate construction of gene graph and imputation of expression levels.

Despite the advantages of G2S3 over the other imputation methods shown in this article, G2S3 can be improved in several directions. First, G2S3 uses a lazy random walk on the gene graph to recover dropout events, i.e., weighted average of the observed expression of the gene of interest and that from neighboring genes. The weights currently depend only on between gene similarity which can be improved by considering the reliability of observed read counts, cell library size, and dispersion of gene expression, similar to the weights used in SAVER. Second, G2S3 does not consider dropout rate and therefore imputes all values at once. This can be improved by calculating the probability of being a dropout for each observed read count and only performing imputation on those with a high dropout probability. Third, we used the MSE criterion for hyperparameter tuning to select the optimal number of steps in G2S3 following a diffusion-based imputation method in a recent study [31]. It should be noted that this is a heuristic approach. Although we performed a real dataset-based validation experiment for this procedure, it is possible that a theoretical approach may give better hyperparameter tuning. Fourth, our model can be further improved by adding two tuning parameters for the second and third terms in the objective function that control the degree of smoothness and sparsity of the resulting gene network. The tuning parameters can be chosen based on the complexity and structure of scRNA-seq data. Finally, G2S3 does not consider the potential subject effect in the data, which has been shown to be prevalent and dominant in certain cell types. One way to address this issue is to consider subject effect as "batch" effect and remove it using batch effect removal tools. This is effective only when there are no other effects of interest confounding the subject effect, for example, disease effect, because they will also be removed together with "batch" effect. When there are other effects that confound with subject

effect and are the interest of study, G2S3 can be improved to consider subject effect and disease effect at the same time in imputation.

## Materials and methods

### G2S3 algorithm

To borrow information from similar genes for data imputation, G2S3 first builds a sparse graph representation of gene network under the assumption that expression levels change smoothly between closely connected genes. Let $X = [x_1, x_2, \ldots, x_m] \in \mathbb{R}^{n \times m}$ denote the observed transcript counts of $m$ genes in $n$ cells, where the column $x_j \in \mathbb{R}^n$ represents the expression vector of gene $j$, for $j = 1, \ldots, m$. We regard each gene $j$ as a vertex $V_j$ in a weighted gene graph $G = (V, E)$, in which the edge between genes $j$ and $k$ is associated with a weight $W_{jk}$.

The gene graph is then determined by the weighted adjacency matrix $W \in \mathbb{R}_+^{m \times m}$. G2S3 searches for a valid adjacency matrix $W$ from the space

$$\mathcal{W} = \{W \in \mathbb{R}_+^{m \times m} : \ W = W^T, \mathrm{diag}(W) = 0\}$$

that is optimal under the assumption of smoothness and sparsity on the graph. To achieve this, we use the objective function adapted from Kalofolias's model [46]:

$$\min_{w \in w} \|W \circ Z\|_{1,1} - 1^T \log(W1) + \frac{1}{2}\|W\|_F^2, \tag{1}$$

where $Z \in \mathbb{R}_+^{m \times m}$ is the pairwise Euclidean distance matrix of genes, defined as $Z_{jk} = \|x_j - x_k\|^2$, $\mathbf{1}$ is a vector of ones, $\|\cdot\|_{1,1}$ is the elementwise L-1 norm, $\circ$ is the Hadamard product, and $\|\cdot\|_F$ is the Frobenius norm. The first term in Eq (1) is equivalent to $2\,\mathrm{tr}(X^T LX)$ that quantifies how smooth the signals are on the graph, where $L$ is the graph Laplacian and tr(.) is the trace of a matrix. This term penalizes edges between distant genes, so it prefers to put a sparse set of edges between the nodes with a small distance in $Z$. The second term in Eq (1) represents the node degree which requires the degree of each gene to be positive to improve the overall connectivity of the gene graph. The third term in Eq (1) controls sparsity to penalize the formation of large edges between genes.

The optimization of Eq (1) can be solved via primal dual techniques [47]. We rewrite Eq (1) as

$$\min_{w \in \omega} \ \mathbb{I}_{\{w \geq 0\}} + 2w^T z - 1^T \log(d) + \|w\|^2, \text{ where } \omega = \left\{w \in R_+^{\frac{m(m-1)}{2}}\right\}, \tag{2}$$

where $w$ and $z$ are vector forms of $W$ and $Z$, respectively; $\mathbb{I}_{\{.\}}$ is the indicator function that takes value 0 when the condition in the brackets is satisfied, infinite otherwise; $d = Kw \in \mathbb{R}^m$ and $K$ is the linear operator that satisfies $W\mathbf{1} = Kw$. After obtaining the optimal $W$, a lazy random walk matrix can be constructed on the graph:

$$M = (D^{-1}W + I)/2, \tag{3}$$

where $D$ is an $m$-dimensional diagonal matrix with $D_{jj} = \sum_k W_{jk}$, the degree of gene $j$, and $I$ is the identity matrix.

The imputed count matrix $X_{\mathrm{imputed}}$ is then obtained by taking a $t$-step random walk on the graph which can be written as

$$X_{\mathrm{imputed}}^T = M^t X^T. \tag{4}$$

By default, G2S3 takes a one-step random walk ($t = 1$) to avoid over-smoothing. Adapted from

a previous study on diffusion-based imputation method [31], we also implement an option of tuning the hyperparameter $t$ based on an objective function that minimizes the MSE between the imputed and observed data, i.e.

$$t^* = \underset{t}{\mathrm{argmin}} \ \|M^t X^T - X^T\|.$$

We assume that a good imputation method is not expected to deviate too far away from the raw data structure in the process of denoising. This criterion enables us to denoise the observed gene expression through attenuating noise due to technical variation while preserving biological structure and variation.

Similar to other diffusion-based methods, G2S3 spreads out counts while keeping the sum constant in the random walk step. This results in the average value of non-zero matrix entry decreasing after imputation. To match the observed expression at the gene level, we rescale the values in $X_{\mathrm{imputed}}$ so that the mean expression of each gene in the imputed data matches that of the observed data. The pseudo-code for G2S3 is given in Algorithm 1.

---

**Algorithm 1:** Pseudo-code of G2S3

1: **Input**: $X$

2: **Result**: $X_{imputed}$ = G2S3($X$)

3: $Z$ = distance($X$)

4: $W = \min_{w \in \mathbb{R}_+^{m(m-1)/2}} \mathbb{I}_{\{w \geq 0\}} + 2w^T z - \mathbf{1}^T \log(d) + \|w\|^2$

5: $D$ = degree($W$)

6: $M = (D^{-1}W + I)/2$

7: $t^* = \mathrm{argmin}_t \ \|M^t X^T - X^T\|$

8: $X_{imputed}^T = M^{t^*} X^T$

9: $X_{rescaled}$ = rescale($X_{imputed}$)

10: $X_{imputed} = X_{rescaled}$

11: **End**

---

## Real datasets

We evaluated and compared the performance of G2S3 and ten existing imputation methods using datasets from eight scRNA-seq studies. Among them, four datasets were generated using the UMI techniques and four were generated by non-UMI-based techniques.

**Reyfman** refers to the scRNA-seq dataset of human lung tissue from healthy transplant donors in Reyfman et al. [23]. The raw data include 33,694 genes and 5,437 cells. To generate the reference dataset, we selected cells with a total number of UMIs greater than 10,000 and genes that have nonzero expression in more than 20% of cells. This resulted in 3,918 genes and 2,457 cells.

**PBMC** refers to human peripheral blood mononuclear cells from a healthy donor stained with TotalSeq-B antibodies generated by the high-throughput droplet-based system [24]. This dataset was downloaded from 10x Genomics website (https://support.10xgenomics.com/single-cell-gene-expression/datasets). The raw data include 33,538 genes and 7,865 cells. To generate the reference dataset, we selected cells with a total number of UMIs greater than 5,000 and genes that have nonzero expression in more than 20% of cells. This resulted in 2,308 genes and 2,081 cells.

**Zeisel** refers to the scRNA-seq dataset of mouse cortex and hippocampus in Zeisel et al. [25]. The raw data include 19,972 genes and 3,005 cells. To generate the reference dataset, we selected cells with a total number of UMIs greater than 10,000 and genes that have nonzero expression in more than 40% of cells. This resulted in 3,529 genes and 1,800 cells.

**Chu** refers to the dataset investigating separation of cell subpopulations in Chu et al. [26]. It measured gene expression of 1,018 cells including undifferentiated H1 and H9 human ES cells and the H1-derived progenitors. The cells were annotated with seven cell subtypes: neuronal progenitor cells (NP), definitive endoderm cells (DE), endothelial cells (EC), trophoblast-like cells (TB), human foreskin fibroblasts (HF), and undifferentiated H1 and H9 human ES cells. We performed preliminary filtering to remove genes expressed in less than 10% of cells, which resulted in 13,829 genes.

**Petropoulos** refers to the dataset studying cell lineage in human embryo development in Petropoulos et al. [27]. It measured expression profiles of 26,178 genes in 1,529 cells from 88 human embryos. Cells were labeled as E3-E7 representing their embryonic day. We performed preliminary filtering to remove genes expressed in less than 5 cells and cells with less than 200 expressed genes. After the filtering, we ended up with 22,934 genes and 1,529 cells.

**Trapnell** refers to the dataset studying the transcriptional dynamics of human myoblasts in Trapnell et al. [28]. scRNA-seq data were collected on undifferentiated primary human myoblasts at time 0 and differentiating myoblasts at 24, 48 and 72 hours. Most of the cells are mature myotubes 72 hours after inducing differentiation. The raw data include 47,192 genes and 372 cells. We performed preliminary filtering to remove genes expressed in less than 10% of cells, which resulted in 13,286 genes.

**Paul** refers to the dataset from a study on the transcriptional differentiation landscape of myeloid progenitors [29]. This dataset includes 3,451 informative genes and 2,730 cells. We used this dataset to evaluate the performance of imputation methods in restoring gene regulatory relationships between well-known regulators.

**Buettner** refers to the dataset in Buettner et al. [30]. This dataset includes mouse ES cells labeled by three cell cycle phases–G1, S, and G2/M via flow sorting. The raw data include 38,390 genes and 288 cells. We used this dataset to evaluate the performance of imputation methods in enhancing gene correlations between periodic marker genes of cell cycle phase. We performed preliminary filtering to remove genes expressed in less than 20% of cells, which resulted in 13,355 genes.

## Performance evaluation

**Expression data recovery.**   We first compared the method performance in recovering gene expression using down-sampled datasets. Down-sampling was performed on three independent UMI-based scRNA-seq datasets (Reyfman, PBMC, and Zeisel) to generate benchmarking observed datasets in a similar framework to previous studies [14,19]. In each dataset, we selected a subset of genes and cells with high expression to be used as the reference dataset and treated them as the true expression. Details on the thresholds chosen to generate the reference datasets are described in the "Real datasets" section. However, unlike previous studies that simulated down-sampled datasets from models with certain distributional assumptions [14] which may incur modeling bias, we performed random binary masking of UMIs in the reference datasets to mimic the inefficient capturing of transcripts in dropout events. The binary masking process masked out each UMI independently with a given probability. In each reference dataset, we randomly masked out 80% of UMIs to create the down-sampled observed dataset.

All imputation methods were applied to each down-sampled dataset to generate imputed data separately. Because imputation methods such as SAVER and MAGIC output the

normalized library size values, we performed library size normalization on all imputed data. We calculated the gene-wise Pearson correlation and cell-wise Spearman correlation between the reference data and the imputed data generated by each imputation method. The correlations were also calculated between the reference data and the observed data without imputation to provide a baseline for comparison. One-sided t-test was used to evaluate whether G2S3 significantly improved the gene-wise and cell-wise correlations compared to other imputation methods. To investigate whether the performance depends on the true expression level, we stratified genes into three categories: widely, mildly, and rarely expressed genes, based on the proportion of cells expressing each gene in the down-sampled observed datasets. Specifically, widely expressed genes are those with non-zero expression in more than 80% of cells, rarely expressed genes are those with non-zero expression in less than 30% of cells, and mildly expressed genes are those that lie in between. The gene-wise and cell-wise correlations in each stratum were used to demonstrate the impact of expression level on the performance of imputation methods.

**Restoration of cell subtype separation.** We applied all imputation methods to the Chu dataset to evaluate their performance in separating different cell types. A good imputation method is expected to stabilize within cell-subtype variation (intra-subtype distance) while maintaining between cell-subtype variation (inter-subtype distance). Principal component analysis was conducted on the raw and imputed data for dimension reduction. We calculated the inter-subtype distance as the Euclidian distance between cells from different cell types, and the intra-subtype distance as the distance between cells of the same cell type, using the top $K$ PCs of the data, for $K = 1,...,50$. The ratio of the average inter-subtype distance to the average intra-subtype distance was used to quantify the performance. The higher this ratio is, the better performance the method has. We also calculated silhouette coefficient, a composite index reflecting both the compactness and separation of different cell types, using the top PCs and the true cell subtype labels. The silhouette coefficient ranges from -1 to 1, with a higher value indicating a better match with the cell subtypes and a value close to zero indicating random clustering [48]. To demonstrate the comparison using cell clustering results, we visualized the raw and imputed data with UMAP plots using the top three PCs and colored cells by the cell subtype labels. The normalized mutual information (MI) and adjusted rand index (RI) were used to measure the consistency between cell clustering results and true cell subtype labels. To demonstrate cell subtype separation based on cell subtype marker genes, we further displayed DE and H1/H9 cells by plotting the log-transformed counts using their marker genes [26]: *GATA6*, a marker gene of DE cells, and *NANOG*, a marker gene of H1/H9 cells.

**Cell trajectory inference.** We assessed the performance of imputation methods in restoring cell trajectory using human preimplantation embryos from different embryonic days in the Petropoulos dataset. We considered the actual embryonic days to represent the true cell differentiation stage or age. Monocle 2 was used to infer pseudo-time from the normalized raw and imputed data [32]. To measure the consistency between the actual embryonic days and the reconstructed pseudo-time, we calculated the pseudotemporal ordering score (POS) and Kendall rank correlation coefficient (Cor). Cell trajectories were visualized by embedding cells into two-dimensional space using reversed graph embedding, a recently developed machine learning method to reconstruct complex single-cell trajectories in the R package Monocle 2 [32].

**Differential expression analysis.** To assess the performance in identifying differentially expressed genes, we compared gene expression between two cell subtypes: H1 and NP cells, using both imputed scRNA-seq and bulk RNA-seq data from the Chu dataset. We also compared gene expression profiles of undifferentiated myoblasts to mature myotubes collected 72 hours after inducing differentiation from the Trapnell dataset. The raw and imputed data were

normalized and log-transformed before evaluation. We used t-test in the bulk RNA-seq data to identify differentially expressed genes and selected the top 200 genes based on P-value as ground truth. We then performed differential expression analysis in the scRNA-seq data using the same test. All the differential expression analysis in the scRNA-seq data was performed using the Seurat R package (version 3.0) with a default threshold to keep genes with at least 1.5-fold change. The prediction accuracy was measured by the area under an ROC curve by comparing the differentially expressed genes identified in the raw and imputed scRNA-seq data at different P-value threshold with ground truth.

**Gene-gene relationship restoration.** We evaluated the method performance by investigating the enhancement in gene regulatory relationships using the Paul dataset and the recovery of gene-gene correlations between periodic marker genes in the Buettner dataset. In gene regulation, a Boolean network constructed by a systematic review on the interactions of core transcription factors to model myeloid differentiation [35] was used as ground truth. The same network was used in the evaluation of DCA [18]. Among the eleven key regulators in the network with known inhibitory and activatory relationships in blood development, ten were present in the Paul dataset. We reconstructed GRN on these ten regulators in the raw and imputed datasets by different methods, using the top four GRN inference algorithms from a review paper [49], PIDC [38], GENIE3 [39], GRNBoost2 [40], and PPCOR [41]. The prediction accuracy of each method was evaluated by comparing the inferred GRN to the ground-truth network using AUROC and AUPRC. The AUROC/AUPRC ratio was calculated by dividing AUROC/AUPRC by that of a random predictor, and the process was repeated for 50 times. The estimated pairwise correlations between genes using the raw unimputed and imputed data by each method were compared for performance evaluation. The Beuttner dataset contains 67 periodic marker genes with peak expression in G1/S and G2/M phases established in a previous study [34]. As marker gene expression varies over cell cycle, we expect pairs of periodic genes whose expression peak during the same cell cycle phase to be positively correlated, and pairs of genes whose expression peak at different phases to be negatively correlated. Pairwise correlations were calculated in the raw and imputed data by each method. The proportion of gene pairs with correct direction of correlation was used to compare the method performance.

## Supporting information

**S1 Fig. Comparison of the mean-variance relationship in gene expression before and after down-sampling.** For each gene, the coefficient of variation (CV) across all cells after down-sampling (y-axis) is plotted against the CV of non-zero cells in the reference data (x-axis). (TIF)

**S2 Fig. Optimal value of hyperparameter in G2S3.** A. Mean squared error (MSE) at different diffusion steps in three down-sampled datasets. B. Gene-wise and cell-wise correlations of G2S3 imputed data at different diffusion steps and the reference data. (TIF)

**S3 Fig. Evaluation of expression data recovery of all imputation methods by down-sampling.** Performance of imputation methods measured by correlation with reference data from the first category of datasets, using gene-wise (top) and cell-wise (bottom) correlation. Box plots show the median (center line), interquartile range (hinges), and 1.5 times the interquartile (whiskers). (TIF)

**S4 Fig. Evaluation of expression data recovery of all imputation methods by down-sampling in three gene strata.** Performance of imputation methods measured by correlation with reference data from the first category of datasets, using gene-wise (top) and cell-wise (bottom) correlation. Genes are stratified into three groups: widely (>80%, left), mildly (30%-80%, middle), and rarely (<30%, right) expressed.
(TIF)

**S5 Fig. Cell subtype marker gene expression in the Chu dataset.** Scatter plot showing expression level of marker genes for DE cells (*GATA6*) and H1/H9 cells (*NANOG*). Cells are colored by the cell subtype labels.
(TIF)

**S6 Fig. Receiver operating characteristic (ROC) curves demonstrating improvement in differential expression analysis in the Chu dataset.** ROC curves measuring the prediction accuracy in scRNA-seq data on differentially expressed genes identified in bulk RNA-seq data comparing H1 to other homogeneous cell types (H1 vs. EC, H1 vs. HF, and H1 vs. TB).
(TIF)

**S7 Fig. Performance of G2S3 in recovering gene regulatory relationships.** Boxplots showing the area under the precision-recall curve (AUPRC) ratios that measure the accuracy of inferred GRNs using the imputed data by different imputation methods. PIDC, GENIE3, GRNBoost2 and PPCOR are used to infer GRNs. Red line indicates the performance of a random predictor.
(TIF)

**S8 Fig. Evaluation of recovering gene correlation relationship of all imputation methods in the Paul dataset.** Heatmaps of pairwise correlations between well-known blood regulators.
(TIF)

**S9 Fig. Expression patterns on four inhibitory gene pairs in the Paul dataset.** Each row shows the scatterplots of a mutually inhibitory gene pair in the raw and imputed data by all methods.
(TIF)

**S10 Fig. Expression patterns on three activatory gene pairs in the Paul dataset.** Each row shows the scatterplots of a mutually activatory gene pair in the raw and imputed data by all methods.
(TIF)

**S1 Table. Comparison of the gene-wise and cell-wise correlations of G2S3 and other methods in down-sampling experiments.** P-values of testing the difference of correlations of G2S3 and other methods with the reference data.
(DOCX)

**S2 Table. Computation time of all imputation methods.** Runtime in minutes for each imputation task using a single processor on an 8-core, 50 GB RAM, Intel Xeon 2.6 GHz CPU machine.
(DOCX)

## Acknowledgments

The authors would like to thank Drs. Smita Krishnaswamy and Hongyu Zhao for their valuable suggestions and comments.

## Author Contributions

**Conceptualization:** Weimiao Wu, Xiting Yan, Zuoheng Wang.

**Data curation:** Weimiao Wu, Qile Dai.

**Formal analysis:** Weimiao Wu, Yunqing Liu.

**Funding acquisition:** Xiting Yan, Zuoheng Wang.

**Investigation:** Weimiao Wu, Xiting Yan, Zuoheng Wang.

**Methodology:** Weimiao Wu, Xiting Yan, Zuoheng Wang.

**Software:** Weimiao Wu.

**Supervision:** Xiting Yan, Zuoheng Wang.

**Visualization:** Weimiao Wu, Yunqing Liu, Xiting Yan, Zuoheng Wang.

**Writing – original draft:** Weimiao Wu.

**Writing – review & editing:** Weimiao Wu, Yunqing Liu, Xiting Yan, Zuoheng Wang.

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
