## [Decision Letter · Decision Letter 0]

17 Jun 2020

Dear Dr. Wang,

Thank you very much for submitting your manuscript "G2S3: a gene graph-based imputation method for single-cell RNA sequencing data" for consideration at PLOS Computational Biology.

As with all papers reviewed by the journal, your manuscript was reviewed by members of the editorial board and by several independent reviewers. In light of the reviews (below this email), we would like to invite the resubmission of a significantly-revised version that takes into account the reviewers' comments.

We cannot make any decision about publication until we have seen the revised manuscript and your response to the reviewers' comments. Your revised manuscript is also likely to be sent to reviewers for further evaluation.

Sincerely,

Qing Nie

Associate Editor

PLOS Computational Biology

Jian Ma

Deputy Editor

PLOS Computational Biology

Reviewer's Responses to Questions

**Comments to the Authors:**

Reviewer #1: Wu et al proposed a new method, G2S3, for imputing single cell gene expression data. The main novelty of this methods is that it borrows information from adjacent genes in a sparse gene graph learned from gene expression profiles across cells. It is a clearly written manuscript and I appreciate the authors’ effort to demonstrate G2S3’s performance on seven single-cell gene expression datasets. However, I think in order to support their central argument that “G2S3 is superior in recovering true expression levels”, more computational and biological evidences are needed, especially since many imputation methods have already been developed for single-cell data. The G2S3 method/model also needs better justification with computational evidences. Please see my detailed comments below.

“Down-sampling was applied to these datasets to assess the method performance in recovering true expression levels.”

Since ground truth gene expression is in general not available for real data, it is acceptable to use the down-sampling approach to create synthetic datasets for assessing method performance. However, it is very important to demonstrate that the down-sampled data really resemble the real data distribution, so that the conclusions on down-sampled data can be generalized to real data. Did the authors filter out any genes or cells in the down-sampling process? Are the gene expression mean-variance relationship and other relationships between key variables preserved in down-sampled data compared to real data.

These questions are also critical since all methods except for G2S3 have poor performance in terms of gene correlation on down-sampled data. Even G2S3’s imputed data does not have a significant better performance than observed data, especially if we look at the worst cases.

For the restoration of cell subtype separation, the authors only compared methods based on 1-10 PCs, which only represent a narrow range. What if the authors look at the results with 15, 20, …, 50 PCs?

Figure 2 compares imputed data based on average inter/intra-subtype distance ratio (top) and silhouette coefficient. Since the true cell types are known, the author should also compare the performance based on cell type clustering/classification. Figure 3 demonstrates the PCA plots for the Chu dataset. It would be helpful to also show the tSNE plots. Another question is whether the plots are based on counts or log-transformed counts?

For the Petropoulos dataset, the authors need to better justify the superior performance of G2S3. First, even though cells from different stages are better separated in G2S3’s imputed data, cells from the same stage are also clustered into several subgroups. The authors need to show that these subgroups are biologically meaningful and are not artifacts of G2S3’s imputation. Second, even though G2S3 and MAGIC both show better results in the previous study, we don’t know if this holds in additional analyses, so the other methods should also be compared on this dataset.

For the study on gene regulatory relationship recovery, the authors argues that “G2S3 showed the greatest enhancement of the regulatory relationship…” However, G2S3 and DCA have very similar performance in Figure 6A. It’s also not clear why “MAGIC performed worse than the raw data”. In additional to plotting the correlation matrices in Figure 6, the authors should also compare the methods in a more strict manner – using a quantitative criterion.

This study is based a few number of gene pairs. The authors should consider using more gene pairs (e.g., known TF-target pairs) for benchmarking.

Methods

(1) G2S3 learns a weighted adjacency matrix for genes using an objective functions adapted from Kalofolias’s model. Since several methods have been developed for single cell data to construct gene networks, what if the authors use any of these methods to learn the adjacency matrix? What’s the additional advantage of model (1)? Since this is the novel part in G2S3’s imputation framework, it is important to demonstrate why model (1) is the best strategy for learn gene-gene relationships.

(2) For the imputation step, why is it best to take one step of the random walk? Like for another diffusion-based methods such as MAGIC, the authors need to justify the selection of this key parameter.

(3) Is any normalization and data transformation methods applied on the count matrix X before applying G2S3?

Reviewer #2: The manuscript entitled “G2S3: a gene graph-based imputation method for single-cell RNA sequencing data” developed a method, G2S3, to impute the dropouts in scRNA-seq data by using the gene graph information. The authors compared G2S3 with several other existing imputation methods with respect to recovering true expression levels, identifying cell subtypes, improving differential expression analyses, and recovering gene correlations. However, the methodology novelty and appropriate benchmarking are not convincingly demonstrated. I will preserve my recommendation until the following concerns can be clearly addressed if the authors are willing to do so.

1. As the authors cited and stated, the key model (Equation 1) was adapted from a previous Kalofolias’s model. The novelty of the proposed method is unclear. In addition, there are three terms in the proposed objective function (Equation 1). In most machine learning studies, two tuning parameters are often added to balance the contribution of each term. It would be better if the authors can add two other tuning parameters and analyze the effect of each term on the results.

2. Now there are a lot of methods that have been proposed for imputing dropout events in scRNA-seq data in the last three years (https://www.scrna-tools.org/tools?sort=name&cats=Imputation). The authors just compared their method with five existing methods. More methods developed recently (e.g., ALRA, EnImpute, knn_smooth, scTSSR, SAUCIE) should be compared.

3. The authors did not consider SAVER in some experiments since, as they explained, it was not able to finish imputation within 24 hours for large scale datasets. In order to compare the proposed method with SAVER, the authors can run the imputation methods on datasets with only selected genes (e.g., highly variable genes).

4. In the differential expression analysis and gene regulatory relationship analysis experiments, only one dataset is used. In order to evaluate the generalization of the proposed methods, more datasets should be considered in these two experiments.

5. It would be better if the authors can test the performance of different methods on other downstream tasks of scRNA-seq analysis (e.g., cell trajectory inference).

6. Fig 1: from my viewpoint, the correlation on the gene level after imputation is too low (~0.5), even just slightly higher than ‘observed’ group. I have two concerns: (1) It was meant to impute dropout (i.e., zeroes), but now it seems that all the expression values were locally averaged. (2) Is it necessary to make a slight improvement on correlation but change the whole expression profile? It’s hard to tell whether an imputation method performs well, just relying on the evaluation of gene expression correlations.

Moreover, Fig 1, left panel: G2S3 median is lower than that of the observed group on Reylman and PBMC datasets, and the variation in G2S3 is higher than observed for all datasets (also for the right panel, cell level correlation). So I don’t agree with the following statement: “In all datasets, G2S3 consistently achieved the highest correlation with the reference data at both gene and cell levels”. The superior of G2S3 was not demonstrated.

7. Fig 3: Why did the authors use PCA but not tSNE or other dimensional reduction methods for scRNA-seq data?

8. Fig 4: the dataset of human embryo stem cells from different embryonic days was used to analyze cell types. But it should be more appropriate to analyze cell differentiation trajectory here. See also the above comment 5.

9. Fig 5: The authors divided the dataset into high-expressed genes and low-expressed genes. But in practice, one would not perform such a step for differential expression (DE) analysis. How about the overall performance of different methods on the whole dataset? For bulk expression data, the authors used T test and Wilcox test for DE analysis. Why not using the standard pipeline of DE analysis, such as DEseq2?

10. Fig 6: it’s hard to tell which method is better for GRN recovering using correlation scatter plot. The results of this part are more descriptive but not quite rigorous.

11. Please use a table to list computation times of all the compared methods for all the analyzed datasets.

12. Page 17, please check whether the following text are typos:

Line 353: “2tr(…)”?

Line 354: “L”?

**Have all data underlying the figures and results presented in the manuscript been provided?**

Reviewer #1: Yes

Reviewer #2: Yes

PLOS authors have the option to publish the peer review history of their article (what does this mean?). If published, this will include your full peer review and any attached files.

Reviewer #1: No

Reviewer #2: No
---

## [Decision Letter · Decision Letter 1]

12 Nov 2020

Dear Dr. Wang,

Thank you very much for submitting your manuscript "G2S3: a gene graph-based imputation method for single-cell RNA sequencing data" (PCOMPBIOL-D-20-00838R1) for consideration at PLOS Computational Biology. As with all papers peer reviewed by the journal, your manuscript was reviewed by members of the editorial board and by several independent peer reviewers. Based on the reports, we regret to inform you that we will not be pursuing this manuscript for publication at PLOS Computational Biology.

Both reviewers still have strong reservation on publishing the manuscript as it currently stands. One major concern is its performance in comparison to many other existing methods and the advantage does not appear to be clear and convincing. Sorry that we could not be more positive but we would be willing to consider a resubmission if the major concerns could be fully addressed.

The reviews are attached below this email, and we hope you will find them helpful if you decide to revise the manuscript for submission elsewhere. We are sorry that we cannot be more positive on this occasion. We very much appreciate your wish to present your work in one of PLOS's Open Access publications. 

Thank you for your support, and we hope that you will consider PLOS Computational Biology for other submissions in the future.

Sincerely,

Qing Nie

Associate Editor

PLOS Computational Biology

Jian Ma

Deputy Editor

PLOS Computational Biology

Reviewer's Responses to Questions

**Comments to the Authors: **

Reviewer #1: The revised manuscript has resolved some of my previous concerns, but I still have the following comments after reading the response letter. In addition, the authors did not include line numbers of the revised contents in the response letter, nor did they highlight/color the revised parts in the updated manuscript, making it difficult to evaluate if some of my previous comments (in particular, comments 1, 6, and 8) are fully addressed. 

For the procedure to filter out genes, the authors applied different sets of thresholds for each dataset. What is the rationale for selecting these thresholds? Some of the thresholds seem to be very stringent and would not be used in real practice. For example, in the Segerstolpe data, the authors used “filtering to remove genes expressed in less than 20% of cells”. 

I still have one concern about Figure 2. Why don’t the authors calculate and report the adjusted rand index or normalized mutual information between inferred clusters and true cell types, both of which are commonly used in single-cell studies to evaluate clustering performance?

“To avoid over-smoothing” is not a solid justification for taking one step of the random walk. I agree that too many steps may result in over-smoothing in imputation, but there should be a more model-guided way to determine the optimal number of steps instead of arbitrarily choosing one.

The authors‘ argument that ” G2S3 is robust to variations in sequencing depth across cells that it performed well in both UMI-based and RPKM datasets used in our experiments.“ is not convincing to me. If G2S3 does not require normalization before imputation, how does it account for the varying sequencing depths in different individual cells? This is necessary for accurate imputation of gene expression levels.

Reviewer #2: In the revised manuscript, the authors have compared G2S3 with several other imputation methods. But unfortunately, G2S3 does not show dominant or convincing superiority in most tasks. For example, in correlation with reference, G2S3 is not significantly higher than scTSSR; in cell trajectory inference, G2S3 is not better than scImpute and VIPER; in differential expression analysis, G2S3 is not better than MAGIC. In addition, Gene-gene relationship recovery is not convincingly demonstrated. 

Below are some specific comments. 

Major: 

1. Fig 1 shows that G2S3 does not perform best, particularly compared to scTSSR which has already considered not only gene similarity but also cell similarity for single cell gene expression imputation. The method novelty and performance improvement of G2S3 is a big problem of my concern. 

The statistical significance for comparisons was not provided, which should be provided at least in the supplementary information. 

2. Fig 5: I don’t think it’s reasonable to divide the whole gene set into highly-expressed genes and lowly-expressed genes, although the authors argued in the response letter that this procedure was for investigating the impact of dropout on imputation (“We divided genes into highly and lowly expressed genes to demonstrate the impact of dropout rate on the imputation method performance”). In my viewpoint, one can perform other more straightforward analysis to directly demonstrate the impact of dropout on the imputation performance, rather than to look at differential expressions. Actually, most literatures of imputation methods rarely divide the gene set into highly-expressed genes and lowly-expressed genes for evaluating and benchmarking their performance in DE analysis. 

The authors also stated that “The high dropout rate of these lowly expressed genes present a harder task for imputation methods to recovery for DE analysis”. If so, does it mean that the imputation methods would perform better on the highly-expressed genes to recovery for DE analysis? But the results in Fig 5 (both A and B, left panels) showed that almost all the methods performed not that well on the highly-expressed genes and even worse than that on the lowly-expressed genes. How to interpret it? 

The results in Fig 5 are not convincing for me to believe G2S3 is a better imputation method for DEG analysis, especially by looking at the results for overall differential expression analysis (Figure A in the response letter). Although the authors emphasized that G2S3 improved DE analysis for the lowly-expressed genes, but it shows much worse performance for the highly-expressed genes. Even for the lowly-expressed genes as the authors emphasized, how to determine a gene is lowly-expressed is also a problem in practice. 

In addition, MAGIC is not used for comparison for the lowly expressed genes (Fig 5A, right panels)? 

3. To evaluate the performance on gene-gene relationship recovery, Table 2 makes some sense but Fig 6 (as well as the related Fig S5-S6) is not very reasonable by examining whether the pair-wise correlation was enhanced since the true correlation is unknow, unless the authors could use, for instance, FISH data for testing (See Fig S3 in Ref [14]). 

Just a little advice: to demonstrate G2S3 could help improve the gene network reconstruction, the authors may considered to perform GRN inference based on the data imputed by G2S3 or other imputation methods and to assess whether the network prediction accuracy (e.g., AUROC or AUPRC) could be increased compared to the raw data. The scRNA-seq data based GRN inference methods as well as the benchmarking dataset can be referred to the following paper:

Pratapa, A., Jalihal, A.P., Law, J.N. et al. Benchmarking algorithms for gene regulatory network inference from single-cell transcriptomic data. Nat Methods 17, 147–154 (2020). https://doi.org/10.1038/s41592-019-0690-6

Minor:

4. Fig 2: It’s hard to distinguish between different curves. A better presentation should be considered. In addition, color-annotation for different methods in Fig 1, Fig 2 and Fig 5 seem not consistent.

**Have all data underlying the figures and results presented in the manuscript been provided?**

Reviewer #1: None

Reviewer #2: Yes

PLOS authors have the option to publish the peer review history of their article (what does this mean?). If published, this will include your full peer review and any attached files.

Reviewer #1: No

Reviewer #2: No

---

## [Decision Letter · Decision Letter 2]

15 Mar 2021

Dear Dr. Wang,

Thank you very much for submitting your manuscript "G2S3: a gene graph-based imputation method for single-cell RNA sequencing data" for consideration at PLOS Computational Biology.

As with all papers reviewed by the journal, your manuscript was reviewed by members of the editorial board and by several independent reviewers. In light of the reviews (below this email), we would like to invite the resubmission of a significantly-revised version that takes into account the reviewers' comments.

We cannot make any decision about publication until we have seen the revised manuscript and your response to the reviewers' comments. Your revised manuscript is also likely to be sent to reviewers for further evaluation.

Sincerely,

Qing Nie

Associate Editor

PLOS Computational Biology

Jian Ma

Deputy Editor

PLOS Computational Biology

Reviewer's Responses to Questions

**Comments to the Authors:**

Reviewer #1: The authors have addressed all my concerns.

Reviewer #2: The authors made effort to revise the manuscript and address my questions. The revision regarding DEG analysis and GRN inference sections makes the performance evaluation more reasonable. The addition of hyperparameter tuning as well as Fig 7 is good. Although the manuscript has been improved, it still has some serious issues which should be clearly addressed.

Major:

1. Methods section.

1.1 “G2S3 algorithm” in the Methods section should be more carefully and clearer described.

For example, “1” in the second term of equation (1) was not explained and it seems that this symbol was not standardly typed—it’s hard to be differentiated from the number 1.

Also, 1{≥0} in equation (2) was not clear. Please refer to ref. 44 for mathematical symbols.

Moreover, “ × ” in equation (4) and the following “ ” (Line 518) are not consistent. Check it also in Algorithm 1.

1.2 Assumptions underlying the objective function of MSE should be described to explain its rationality used for selecting optimal step parameter.

1.3 I would suggest to move “Hyperparameter tuning in G2S3” in the Results section to the Methods section (after “G2S3 algorithm”), to make it more coherent. Also, Fig S2 could be moved to the main text, as it would strengthen the method part of this manuscript.

2. Differential expression analysis evaluation section (Fig 5). Although the results presented here show better performance for G2S3, but I have some concerns.

2.1 In the previous versions of the manuscript (including R1 version), Segerstolpe dataset was used for benchmarking. But in this revised version (R2), this dataset was changed to Trapnell dataset. Why? It seems that the authors only artificially selected results that are favorable for G2S3.

2.2 For the Chu dataset, in the previous versions (R1), the DEGs were analyzed by comparing DE and H1 cells. However, in the current version (R2), the DEGs were for NP vs H1 cells. This is also reflected in Line 547 (Methods section). I have the same concern with the above comment that the authors only artificially selected results that are favorable for G2S3.

To address the above issues, the authors should perform a more justified evaluation and comparison of DEG analysis. I would like to see new results on more comprehensive datasets and cell types as mentioned above.

2.3 How ROCs were generated is not clear. Did the authors use fold changes of genes in scRNAseq as score to be compared with “ground truth” (i.e., top 200 DEGs in bulk RNA-seq), for calculating ROC?

3. GRN inference section (Fig 6). The improvement of G2S3 in GRN inference is an impressive result in this manuscript. I have the following questions and suggestions.

3.1 The authors used Paul dataset to infer GRN. The ground truth of the GRN was not listed out so it’s not clear how they evaluated the network prediction accuracy. The Ref 29 only provides gene targets of 4 TFs (in its supplementary materials). How did the authors use this information as ground truth? How many genes in the Paul dataset were used to reconstruct GRN? How they evaluated the prediction accuracy? Details are needed.

3.2 The author only used GENIE3 and PROC for evaluation. However, there are many other methods for network reconstruction. At least, PIDC, GENIE3 and GRNBoost2 should be included, which have been shown as top performing method (Nat Methods 17, 147–154 (2020). https://doi.org/10.1038/s41592-019-0690-6). The addition of these methods for evaluation would be beneficial or the readers to assess which GRN method could be used in combination with G2S3 imputation.

3.3 In addition to AUPRC, AUROC is also commonly used for network prediction evaluation and thus is desired here.

Minor:

1. Abstract section. “novel” should be avoid used here, since the network learning algorithm, the major part of G2S3, is just adapted from Ref. 44. Lines 20-23: should be revised for “overall performance”.

2. Introduction section. Line 88 “… is robust to outliers in the data” is not verified.

3. Reference format should be carefully checked.

4. Fig S3 seems almost repeated with Fig 1, only with additional SAUCIE.

5. Fig S4. X-label, full text of “Widely (Mildly/Rarely) expressed genes” may be better.

6. Fig 4 legend: “reverse graph embedding” is not clear. R packages or functions used for visualization here should be provided.

7. The phrase “recovering true gene expression” occurred many times throughout the manuscript but I don’t think this phrase is accurate. Not only the dropout but also the other expression values were globally changed after imputation (see Lines 520-521). So the recovered gene expression is not the true expression.

8. “cell lines” (Line 185 and other places) is not accurate. Cell line means in vitro cultured cells, but undifferentiated human ES (H1 and H9) should be from in vivo tissues.

9. Page 20 Lines 421-422, “especially for genes with relatively low expression” seems a mistake expression? In addition, “G2S3 is the most computationally efficient…” seems not true, which is not reflected in Table S2.

10. Page 29, “Improvement in …”, this is repeated with that in the Results section. I suggest to change such words in the methods section to, like, “Evaluation on …”.

11. English should be improved. The text is not concise enough and sometimes not accessible. Grammar errors even still exist.

Just to list a few:

Fig 5 legend;

Line 76;

Lines 82-83;

Line 296;

Line 545. …

**Have all data underlying the figures and results presented in the manuscript been provided?**

Reviewer #1: None

Reviewer #2: None

PLOS authors have the option to publish the peer review history of their article (what does this mean?). If published, this will include your full peer review and any attached files.

Reviewer #1: No

Reviewer #2: No
---

## [Decision Letter · Decision Letter 3]

29 Apr 2021

Dear Dr. Wang,

We are pleased to inform you that your manuscript 'G2S3: A gene graph-based imputation method for single-cell RNA sequencing data' has been provisionally accepted for publication in PLOS Computational Biology.

Best regards,

Qing Nie

Associate Editor

PLOS Computational Biology

Jian Ma

Deputy Editor

PLOS Computational Biology

Reviewer's Responses to Questions

**Comments to the Authors:**

Reviewer #2: The authors have addressed my questions. Now the revised manuscript is suitable to be published.

**Have the authors made all data and (if applicable) computational code underlying the findings in their manuscript fully available?**

Reviewer #2: Yes

PLOS authors have the option to publish the peer review history of their article (what does this mean?). If published, this will include your full peer review and any attached files.

Reviewer #2: No

---

## [Editor Report · Acceptance letter]

14 May 2021

PCOMPBIOL-D-20-00838R3 

G2S3: A gene graph-based imputation method for single-cell RNA sequencing data

Dear Dr Wang,

I am pleased to inform you that your manuscript has been formally accepted for publication in PLOS Computational Biology. Your manuscript is now with our production department and you will be notified of the publication date in due course.

With kind regards,

Zsofi Zombor
